# Position Estimation of Obstacles Using Maximum Expectation Stochastic Gradient Variational Inference in Intelligent Control Game Systems

Xiangri Lu[a,b,*]

[a]Beijing Institute of Technology, Haidian District, Beijing, PRC

[b]National Key Lab of Autonomous Intelligent Unmanned Systems, Haidian District, Beijing, PRC

**Abstract**

This paper discusses the optimization of the confrontation position of agent in a two-dimensional game system with incomplete information, based on the random movements of agent relative to obstacles. Incomplete information games contain numerous unknown factors, adding complexity to intelligent control game systems. To address these unknown factors, intelligent control game systems often require substantial data and significant computational resources. However, real confrontation systems involve intelligence gathering, and neither side will have a complete information set about the confrontation situation. To tackle this issue, this paper proposes a Maximum Expected Stochastic Gradient Variational Inference Algorithm within the Q-learning framework, which can infer the position coordinates of obstacles in the confrontation plane. In this paper, the experimental data of Q-learning model of reinforcement learning，The Maximum Expectation Stochastic Gradient Variational Inference algorithm is then used to estimate the position coordinates of the obstacles arranged by player B.

**Keywords:** Incomplete Information; Intelligent Control Game; Q-learning; Maximum Expected Random Gradient Variational Inference; Reinforcement Learning

1.Introduction

In 1977, Arthur P. Dempster, Nan Laird, and Donald Rubin introduced the Expectation-Maximization (EM) algorithm in their seminal paper, establishing it as a cornerstone in statistical modeling and data analysis[1-3]. This algorithm is particularly valuable for finding local maximum likelihood parameters of statistical models in situations where equations cannot be directly solved, enabling the estimation of unobservable data. The iterative nature of the EM algorithm consists of

two alternating steps: Expectation (E) and Maximization (M). The process of alternating E and M steps continues until the algorithm converges to a set of parameters that maximizes the likelihood function.This iterative refinement allows the EM algorithm to effectively estimate parameters in the presence of incomplete data.

When it comes to optimizing the confrontation position of an agent in a two-dimensional game system with incomplete information,the expectation-maximization algorithm also has some drawbacks.First, the convergence speed of the algorithm can be slow, especially in cases where the data has high dimensionality or the initial estimates are far from the optimal parameters. Each iteration includes both the Expectation and Maximization steps, which are computationally expensive. This makes the EM algorithm less suitable for very large-scale datasets or applications with strict time constraints.

In 2013, Matthew D. Hoffman and Andrew Gelman introduced the Stochastic Gradient Variational Inference (SGVI) algorithm, which has since become a vital method for handling complex probabilistic models through approximating posterior distributions[4,5]. The SGVI algorithm combines the efficiency of stochastic optimization with the flexibility of variational inference, making it highly suitable for large-scale data problems and high-dimensional models. Unlike traditional inference methods, SGVI iteratively updates model parameters using stochastic gradient descent, allowing it to effectively scale to scenarios with incomplete information and dynamic environments.The principles and functionalities of SGVI make it particularly relevant in fields such as game theory and warfare simulation. In game theory, SGVI can be used to optimize strategies in competitive scenarios where information is often incomplete and constantly changing. Similarly, in warfare simulation, SGVI helps optimize the positioning of agents in dynamic, uncertain environments, improving decision-making and strategic planning.

Firstly, the SGVI algorithm utilizes mini-batch data for updates, allowing it to effectively handle large-scale datasets without needing to process the entire dataset in each iteration. This significantly reduces the consumption of computational resources. Secondly, since SGVI uses stochastic gradient descent in each iteration, it adapts well to high-dimensional models. Traditional variational inference methods may struggle

with high-dimensional scenarios, whereas SGVI remains efficient in such environments. Lastly, SGVI excels in dealing with missing or incomplete data. By estimating the posterior distribution of hidden variables through variational inference, it enables effective inference even in the presence of incomplete data.

In the study of incomplete information game systems on a two-dimensional plane, agents need to optimize their positions based on the random movement of obstacles. The incomplete information game includes many unknown factors, increasing the complexity of intelligent control game systems. To effectively address these unknown factors, this paper combines the Expectation-Maximization (EM) algorithm and the Stochastic Gradient Variational Inference (SGVI) algorithm. This combination aims to improve the efficiency and accuracy of the agents' decision-making processes.

By integrating the EM algorithm with the SGVI algorithm, we can fully exploit the strengths of both methods in intelligent control game systems. The EM algorithm provides a robust tool for handling hidden variables and incomplete data, while the SGVI algorithm enhances the model's scalability and real-time performance through efficient stochastic gradient updates. Specifically, the EM algorithm can be employed for initial parameter estimation and handling hidden variables, whereas the SGVI algorithm optimizes model parameters in each iteration, utilizing small batches of effective data to support the agents' decision-making process.

This combined approach significantly reduces reliance on large-scale data and massive computational resources while improving the precision and efficiency of position optimization. Ultimately, this method allows intelligent control game systems to achieve optimal decision-making for agents in complex environments with incomplete information.

2 Reinforcement Learning Framework and Algorithms

Assuming that the two units are playing games on the two-dimensional plane, let's suppose that there is the player A, and the player B sets up obstacles at two points (which can be ignored in terms of size) to construct each other. In a $x_{unit} \times y_{unit}$ area( $x_{unit}$ and $y_{unit}$ are each a unit length arbitrarily.)Construct a two-dimensional game environment in the plane. This article sets the starting point of

the player A as the red mark point, and the player A looks for the base camp b of player B, and the player B that sets up obstacles to prevent Player A from attacking the base camp b of player B is marked in green, and the base camp b of player B which is a fixed point $(x_{goal}, y_{goal})$ in the two-dimensional game plane is marked in blue. At this point, This article construct a model that a game triangle in the two-dimensional game plane with the player A and the obstacles arranged by the player B and the base camp b of player B , as shown in Figure 1.

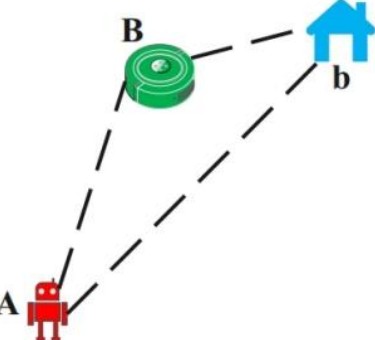

Figure 1 The Composition of Game Participants in a Two-dimensional Game Plane Forms a Game Triangle.

2.1 Reinforcement Learning Framework

In this paper, the player A adopts the reinforcement learning model of the value function Q-learning method[6-8]. The player A interacts with the confrontational game environment, and The player A learns from the environment how to adapt to the current environment to complete the task. The simplified diagram of the reinforcement learning scheme in this article is shown in 3. When the player A completes a certain task, it first interacts with the dynamic adversarial game environment through action $a_t$ . Under the action $a_t$ of the action and the dynamic environment of the confrontation game, the player A will generate a new state. At the same time, the player A reward value is given under the action of the dynamic environment of the confrontation game environment. If this iteration continues, the continuous interaction between the player A and the confrontational game environment will generate a large amount of data. The player A uses the data generated by the reinforcement learning algorithm to modify its own action strategy, and then interacts with the confrontation game environment to generate new data, and uses the new data to further improve own behavior of the player A. After several

generations of learning, , the player A finally learns the best position which is the optimal location weaken the base camp b of player B to weaken the base camp b of player B , that is, the command system when the player A and the player B are in confrontation.

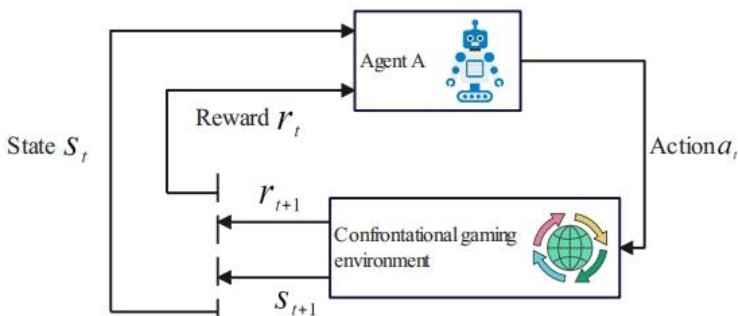

Figure 2 A Simplified Diagram of the Confrontation Game Scheme Based on Reinforcement Learning

2.1.1 Definitions Related to Reinforcement Learning Models

Ⅰ.Definition

According to the above description of player of the confrontation game, the state of the agent in the two-dimensional plane of the confrontation game can be defined:

(1) If the player A encounters an obstacle set by the player B during reinforcement training, the reward value obtained is -1.

(2) If the player A encounters the base camp b of player B, then the reward value obtained is +1.

(3) When the player A is trained in the reinforcement learning process in the blank area of the two-dimensional plane, if the distance between the player A and the base camp b of player B on the plane is within the radius range without any obstacles set by the player B, the confrontation environment will reward the player A with a value of 10. To simplify the problem, we assume that the number of obstacles set by the player B is 1.

(4) The action of the player A is defined as: up (U), down (D), left (L), and right (R).

According to the above definition, the movement rules of the player A in the confrontation game are: action = 0, 1, 2, and 3 respectively represent the movement of the player A in the confrontation game in the four directions of upward, downward, left, and right.

Based on the above the player A experimental environment argument and Figure 2, a simplified diagram based on the reinforcement learning in the confrontation game, we can now draw a state transition running diagram for the player A, as shown in Figure 3.

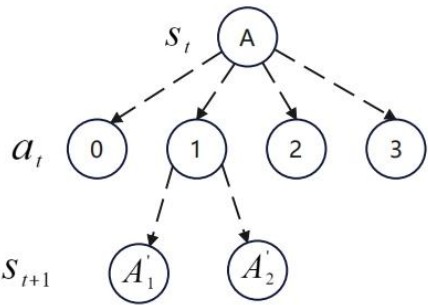

Figure 3 State Transition Diagram of the player A Operation

The two subjects of this reinforcement learning model are the player A, and the two-dimensional planar environment of the confrontation game (the player A, the obstacles arranged by the player B and the base camp b of player B). Under the framework of the Markov Decision Process[9-11] , the player A completed with the support of five key elements: state set, action set, reward set, policy probability distribution and state transition probability distribution closed-loop cycle in reinforcement learning environment. Then let the state set be $S = \{Aggressive\ Offensive,\ Passive\ Offensive\}$ . The set of actions is $a = \{Up, Down, Left, Right\}$ . The reward set is $R = \{-1, 0, 1, 10\}$ .

Ⅱ.The algorithm of secondary $\varepsilon - greedy$ strategy

1.Action choice

And randomly generated a $r$ that floating point number between 0 and 1;if $r < \varepsilon$ ,then exploratory selection is performed, i.e. one action is randomly selected from all actions；if $\varepsilon \leq r \leq 1$ ,Make $Q_{\max}$ to select greedy $a_t$ that in the current $Q$ table， so Select the optimal action $a = \arg\max_{a'} Q(s, a')$ , Select an action with a value $Q_{\max}$ in the current state $s$ ,If $Q_{\max}$ is due to the action selected in $action = \{right, up\}$ ,Then the agent selects $a = \arg\max_{a'} Q(s, a')$ from the action set to continue to explore,If $Q_{\max}$ is due to the action selected in $action = \{left, down\}$ ,Then the agent randomly selects one action from all the actions.

2. $\varepsilon - greedy$ process

First initialize the Q table.Second, repeat the following steps until the maximum

cycle number or convergence is reached：In a given state, one action is selected based on two $\varepsilon - greedy$ strategies[12-14];Perform the action and observe the feedback of the environment and the next state $s'$ .Then update the Q table.Eventually update the status to $s'$ .The Python simulation program of $\varepsilon - greedy$ process for player A is presented in Figure 4.

**Algorithm 1** Design Program for The Player A
  **Constant:** $MAZE\_R = y_{unit}$ , $MAZE\_C = x_{unit}$
1: **function** INITIALIZE$(\alpha, \gamma)$
2:   $states \leftarrow [0, 1, ..., MAZE\_R \times MAZE\_C - 1]$
3:   $actions \leftarrow ['u', 'd', 'l', 'r']$
4:   Initialization $rewards$ 和 $q\_table$
5: **end function**
6: **function** CHOOSEACTION$(state, \epsilon)$
7:   **if** $random(0, 1) > \epsilon$ **then**
8:    **return** Randomly select a valid action.
9:   **else**
10:    **return** Select the action with the highest Q-value from the effective actions.
11:   **end if**
12: **end function**
13: **function** UPDATEQVALUE$(state, action, reward, next\_state)$
14:   $q\_table[state, action] \leftarrow q\_table[state, action] + \alpha \times ($
    $reward + \gamma \times \max(q\_table[next\_state]) - q\_table[state, action])$
15: **end function**
16: **function** GETVALIDACTIONS$(state)$
17:   According to $state$ Valid Action Set Returned by Location
18: **end function**
19: **function** GETNEXTSTATE$(state, action)$
20:   Based on the current $state$ And $action$ Calculate and return
   $next\_state$
21: **end function**

Figure 4 The player A Simulation Test Process

2.1.2 Q-learning incorporating the EM-SGVI algorithm

In the context of war game simulations, intelligent control systems often face challenges due to incomplete information about an opponent's strategy and location. The EM-SGVI algorithm addresses these challenges by iteratively refining estimates of hidden variables, such as obstacle locations, enabling the system to make more informed decisions based on the probability distribution of unknown factors.

When the EM-SGVI algorithm is integrated into the Q-learning framework, it enhances the agent's ability to optimize its behavior in a 2D game environment. Q-learning is a reinforcement learning technique that enables the agent to learn the optimal policy by interacting with the environment and receiving rewards. The EM-SGVI algorithm's ability to estimate and adjust for the presence of obstacles in the environment further improves the agent's ability to identify and occupy the

optimal adversarial position.

The EM-SGVI algorithms and Q-learning framework are utilized to train player A in navigating and weakening the base camp b of player B on a 2D plane map. The process structure is illustrated in Figure 5.

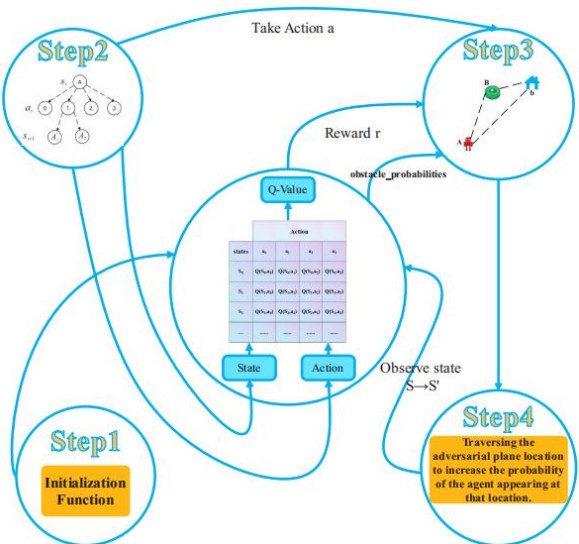

Figure 5 Process Flowchart of the player A Simulation Results

The distribution of obstacle locations set by player B in the reinforcement learning model environment will be analyzed using EM-SGVI. Next, we will provide a convergence proof for the EM-SGVI method.

**Proof:** Suppose X is the known data of the particle unit of player A, that is, the observation variable; Z is the hidden variable, that is, player B arranges the distribution position information of the obstacle unit, then

$$\frac{p(X \mid Z)p(Z)}{p(X)} = p(Z \mid X) \qquad (1)$$

$p(X \mid Z)$ indicates the probability that player A gets the best position if he knows where player B places the obstacle, $p(Z)$ represents the assumption of the distribution of player B arrangement barriers for the player A construct, $p(X)$ is the integration of the position probability against player A, $p(Z \mid X)$ That is, the posterior distribution of the probability that player B arranges obstacles in some positions when player A navigates a point in the confrontation game plane. The final expression process of player A is as follows, so player A is selected as (16).

$$\log p(x) = \log p(x,z) - \log p(z \mid x) \qquad (2)$$

$$\log p(x) = \log p(X, Z) - \log p(Z \mid X)$$
$$= \log \frac{p(X, Z)}{q(Z)} - \log \frac{p(Z \mid X)}{q(Z)} \tag{3}$$

$$left = \int_Z \log p(X) q(Z) dZ = \log p(X) \int_Z q(Z) dZ = \log p(X) \tag{4}$$

$$right = \int_Z \log \frac{p(X, Z)}{q(Z)} q(Z) dZ - \int_Z \log \frac{p(Z \mid X)}{q(Z)} q(Z) dZ \tag{5}$$
$$= ELBO + KL(q(Z) \| p(Z \mid X))$$

$$\log p(x) = ELBO + KL(q(Z) \| p(Z \mid X)) \tag{6}$$

$$\log p(x) \geq ELBO(iff . KL(q(Z) \| p(Z \mid X)) \geq 0) \tag{7}$$

$$Set \quad ELBO = E_{q_\varphi(Z)}[\log \frac{p_\theta(x^{(i)}, z)}{q_\varphi(Z)}] = E_{q_\varphi(Z)}[\log p_\theta(x^{(i)}, z) - \log q_\varphi(Z)] = L(\varphi)$$

$$\hat{\varphi} = \underset{\varphi}{\arg\max} \, L(\varphi) \tag{8}$$

$$\nabla_\varphi L(\varphi) = \nabla_\varphi E_{q_\varphi(z)}[\log p_\theta(x^{(i)}, z) - \log q_\varphi(z)]$$
$$= \nabla_\varphi[\int q_\varphi(z) \log p_\theta(x^{(i)}, z) dz - \int q_\varphi(z) \log q_\varphi(z) dz] \tag{9}$$
$$= \underbrace{\int \nabla_\varphi q_\varphi(z)[\log p_\theta(x^{(i)}, z) - \log q_\varphi(z)] dz}_{\alpha} + \underbrace{\int q_\varphi(z) \nabla_\varphi[\log p_\theta(x^{(i)}, z) - \log q_\varphi(z)] dz}_{\beta}$$

$$\beta = \int q_\varphi(z) \nabla_\varphi[\log p_\theta(x^{(i)}, z) - \log q_\varphi(z)] dz$$
$$= -\int q_\varphi(z) \nabla_\varphi \log q_\varphi(z) dz$$
$$= -\int q_\varphi(z) \frac{1}{q_\varphi(z)} \nabla_\varphi q_\varphi(z) dz \tag{10}$$
$$= -\nabla_\varphi \int q_\varphi(z) dz = 0$$

$$\alpha = \int \nabla_\varphi q_\varphi(z)[\log p_\theta(x^{(i)}, z) - \log q_\varphi(z)] dz$$
$$= \int q_\varphi(z) \nabla_\varphi \log q_\varphi(z)[\log p_\theta(x^{(i)}, z) - \log q_\varphi(z)] dz \tag{11}$$
$$= E_{q_\varphi(z)}[\nabla_\varphi \log q_\varphi(z)[\log p_\theta(x^{(i)}, z) - \log q_\varphi(z)]]$$
$$\therefore \nabla_\varphi L(\varphi) = E_{q_\varphi(z)}[\nabla_\varphi \log q_\varphi(z)[\log p_\theta(x^{(i)}, z) - \log q_\varphi(z)]]$$

If player B arranges the obstacle distribution position information hidden variable Z can simulate the position information several times, that is $Z^{(l)} \sim q_\varphi(z), l = 1, 2, ..., l$ , so

$$\nabla_\varphi L(\varphi) = \frac{1}{L} \sum_{l=1}^{L} \nabla_\varphi \log q_\varphi(z^{(l)})[\log p_\theta(x^{(i)}, z^{(l)}) - \log q_\varphi(z^{(l)})] \tag{12}$$

Suppose that the location information distribution of player B arrangement of obstacles is $z = g_\varphi(\varepsilon, x^{(i)}), \varepsilon \sim p(\varepsilon)$ , $z \sim q_\varphi(z \mid x^{(i)})$ 。

$$\therefore \nabla_\varphi L(\varphi) = \nabla_\varphi E_{q_\varphi(z)}[\log p_\theta(x^{(i)}, z) - \log q_\varphi(z)]$$
$$= \nabla_\varphi \int q_\varphi(z)[\log p_\theta(x^{(i)}, z) - \log q_\varphi(z)]dz$$
$$= \nabla_\varphi \int_\varepsilon [\log p_\theta(x^{(i)}, z) - \log q_\varphi(z)]p(\varepsilon)d\varepsilon$$
$$= \nabla_\varphi E_{p(\varepsilon)}[\log p_\theta(x^{(i)}, z) - \log q_\varphi(z)]$$
$$= E_{p(\varepsilon)}[\nabla_\varphi[\log p_\theta(x^{(i)}, z) - \log q_\varphi(z)]]$$
$$= E_{p(\varepsilon)}[\nabla_z[\log p_\theta(x^{(i)}, z) - \log q_\varphi(z \mid x^{(i)})] \cdot \nabla_\varphi Z]$$
$$= E_{p(\varepsilon)}[\nabla_z[\log p_\theta(x^{(i)}, z) - \log q_\varphi(z \mid x^{(i)})] \cdot \nabla_\varphi g_\varphi(\varepsilon, x^{(i)})] \qquad (13)$$

$\varepsilon^{(l)} \sim p(\varepsilon), l = 1, 2, ..., l$

$$\therefore \nabla_\varphi L(\varphi) \approx \frac{1}{L} \sum_{l=1}^{L} \nabla_{z=g_\varphi(\varepsilon, x^{(i)})}[\log p_\theta(x^{(i)}, g_\varphi(\varepsilon^{(l)}, x^{(i)})) - \log q_\varphi(g_\varphi(\varepsilon, x^{(i)}) \mid x^{(i)})] \cdot \nabla_\varphi g_\varphi(\varepsilon, x^{(i)}) \qquad (14)$$

$$\therefore \varphi^{(t+1)} \leftarrow \varphi^{(t)} + \lambda^{(t)} \nabla_\varphi L(\varphi) \qquad (15)$$

If Q-learning the greedy strategy distribution,Substituting $\pi(s_{t+1}) = \arg\max_a Q(s_{t+1}, a)$ into the Equation (15),then(16).

$$a^{t+1} \leftarrow a^t + \lambda^{(t)} \nabla_a Q(S_{t+1}, a) \qquad (16)$$

when $a^{t+1} = a^t$ ,It means that player A gets the best attack position.

3   Experiment
3.1 Experimental Simulation Platform Environment

The the player planning simulation hardware platform used in this article is equipped with an Intel(R) Core(TM) i7-10750H CPU @ 2.60GHz processor and an NVIDIA GeForce GTX 1660 Ti (6.0 GB) graphics card. Based on this hardware system, we use Python 3 to implement the player and the confrontation game process with incomplete information, and finally enter the program editing interface by "Jupyter Notebook" in the terminal.

3.2 Experimental result

This experiment conducted a comparative study between the improved EM-SGVI algorithm and the Q-learning algorithm. Through 1000 iterations of testing, the experiment was analyzed from two dimensions: runtime and the spatial

distribution of obstacles by player B.Comparative analysis of the performance of EM-SGVI and Basic Q-Learning algorithm,as shown in table 1.

Table 1 Comparative analysis of the performance of Table 1 EM-SGVI and Basic Q-Learning algorithm

| Contrast the project | Statistical indicators | EM-SGVI | Basic Q-Learning | Difference |
|---|---|---|---|---|
| runtime | Average (seconds) | 0.0448 | 0.0404 | +0.0044 |
| | Standard deviation (seconds) | 0.0069 | 0.0051 | +0.0018 |
| | coefficient of variation | 15.40% | 12.62% | +2.78% |
| Obobstacle and target distance | Average (seconds) | 2.2738 | 4.4343 | -2.1605 |
| | Standard deviation (seconds) | 1.0922 | 1.8946 | -0.8024 |
| | coefficient of variation | 48.03% | 42.73% | +5.30% |

The results indicated that the EM-SGVI algorithm showed significant advantages in predicting obstacle positions. The average distance between obstacles and target points for the EM-SGVI algorithm (2.2738) was reduced by 48.72% compared to the Q-learning algorithm (4.4343). This result demonstrates the better adaptability of the EM-SGVI algorithm in adversarial game environment planning, as shown in Figure 6.

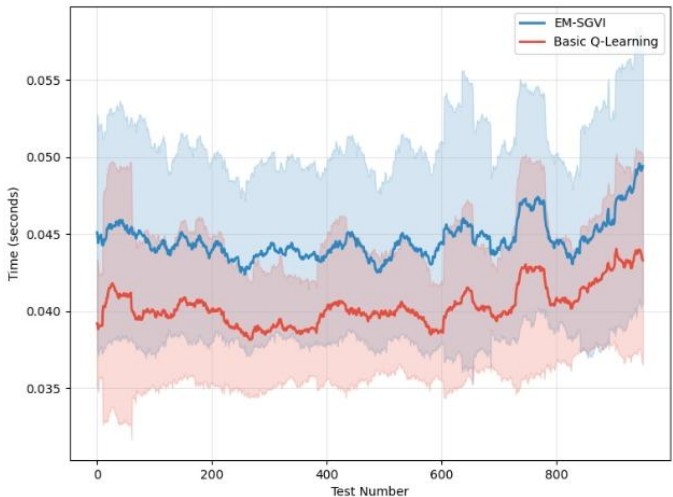

Figure 6    Comparison of Spatial Distance Distribution Between Obstacles and Base Camp for Basic Q-Learning and EM-SGVI Algorithms

From the perspective of stability, the standard deviation of the distance for the EM-SGVI (1.0922) was much lower than that of the basic Q-learning (1.8946), indicating higher predictability and consistency in the spatial layout of obstacles. In terms of computational efficiency, the average runtime of the EM-SGVI algorithm (0.0448 seconds) was only 10.89% higher than the basic Q-learning (0.0404 seconds). This slight increase in time overhead is acceptable, especially considering the significant spatial optimization effects it brings. The difference in the standard deviation of time (0.0069 seconds for EM-SGVI and 0.0051 seconds for basic Q-learning) suggests that both algorithms have good time stability, as shown in Figure 7.

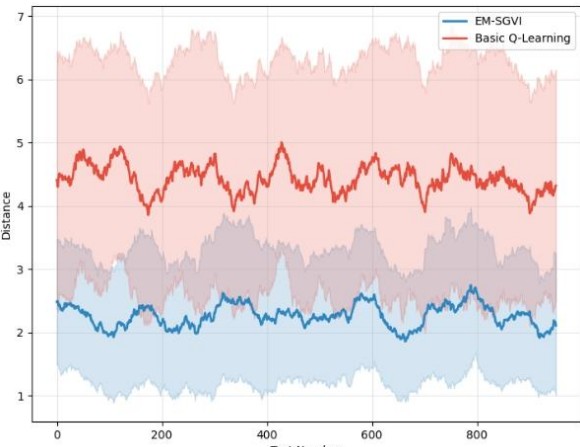

Figure 7 Comparison of Single Cycle Runtime Between Basic Q-Learning and EM-SGVI Algorithms

Heatmap analysis visually shows that the EM-SGVI algorithm exhibits more concentrated and strategic distribution characteristics in obstacle position selection, whereas the basic Q-learning presents a relatively uniform random distribution. These experimental data strongly prove that the EM-SGVI algorithm can significantly improve the accuracy of obstacle position prediction while maintaining computational efficiency, as shown in Figure 8 and Figure 9.

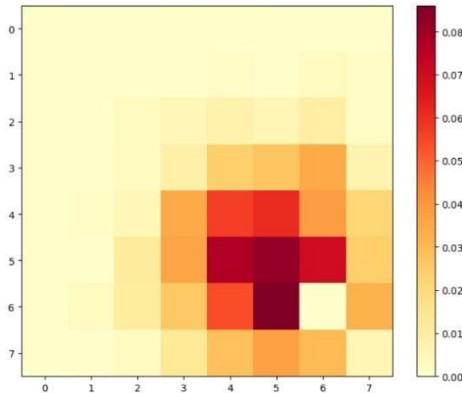

Figure 8 Heatmap of Obstacle Position Estimation by the EM-SGVI Algorithm

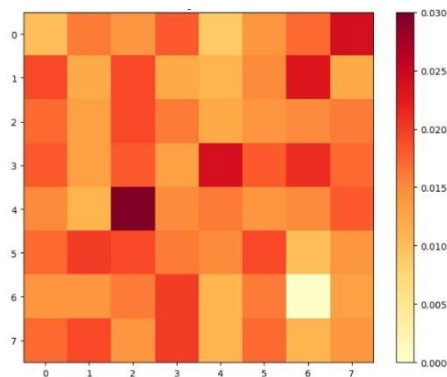

Figure 9 Heatmap of Obstacle Position Estimation by the Basic Q-Learning Algorithm

4.Conclusion

In the study of incomplete information game systems on a two-dimensional plane, agents need to optimize their positions based on the random movement of obstacles. The presence of unknown factors increases the complexity of these intelligent control game systems. To effectively address these challenges, this paper combines the Expectation-Maximization (EM) algorithm with the Stochastic Gradient Variational Inference (SGVI) algorithm to enhance the efficiency and accuracy of agents' decision-making processes.

The integration of the EM algorithm and the SGVI algorithm leverages the strengths of both methods. The EM algorithm handles hidden variables and incomplete data, while the SGVI algorithm improves the model's scalability and real-time performance through efficient stochastic gradient updates.

This combined approach reduces the reliance on large-scale data and massive computational resources, while enhancing the precision and efficiency of position optimization. Ultimately, it enables intelligent control game systems to achieve

optimal decision-making for agents in complex environments with incomplete information.

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
