# OpenReview forum: "Position Estimation of Obstacles Using Maximum Expectation Stochastic Gradient Variational Inference in Intelligent Control Game Systems"
_mathai.club/MathAI/2025/Conference — MathAI 2025 Oral_

### Official Review · Reviewer_RB9C · 2025-02-26
**Review for Position Estimation of Obstacles Using Maximum Expectation Stochastic Gradient Variational Inference in Intelligent Control Game Systems**

**Rating:** 5
**Confidence:** 3

**Review:**

The author proposes to combine two algorithms, namely, the Expectation-Maximization (EM) algorithm and the Stochastic Gradient Variational Inference (SGVI) algorithm to solve the incomplete information game systems on a two-dimensional plane. The author provides mathematical proofs and experimental results to show the efficiency of his method. However, the paper is poorly written. Moreover, the experiments are vague. The paper can use more context and maybe study the results of previous methods.

---

### Official Review · Reviewer_xEcV · 2025-02-28
**The paper integrates two established algorithms, EM and SGVI, which is innovative.**

**Rating:** 7
**Confidence:** 4

**Review:**

The paper integrates two established algorithms, EM and SGVI, which is innovative. Combining EM's ability to handle hidden variables with SGVI's efficiency in stochastic optimization could offer better performance. The application in a reinforcement learning context, specifically Q-learning, is practical for real-world scenarios like autonomous systems or game AI. The experimental results show a significant reduction in obstacle-to-target distance, which indicates effectiveness. Also, the computational overhead is minimal compared to the benefits, which is a plus for real-time applications.

The paper mentions using EM and SGVI but could elaborate more on how exactly they are integrated. The equations are presented, but the explanation might be too dense for readers not familiar with variational inference. The experiment section compares EM-SGVI with basic Q-learning, but it's unclear if other state-of-the-art methods were considered.

Final, the paper presents a promising approach for obstacle estimation in incomplete-information environments, balancing computational efficiency and accuracy. Addressing technical clarity, expanding comparisons, and incorporating trustworthiness mechanisms will strengthen its impact in both academic and industrial AI applications.

---

### Decision · Program_Chairs · 2025-03-08

**Decision:**

Accept (Oral)

**Comment:**

Your article has been accepted and you can make a presentation on the article. All articles will be sorted by rating and within the available conference places one author from each article will be invited. If there are not enough places, then you will either have the opportunity to present remotely or come at your own expense!